# Reaching the Challenging Diagnosis of Complicated Liver Hydatid Disease: A Single Institution’s Experience from an Endemic Area

**DOI:** 10.3390/medicina57111210

**Published:** 2021-11-05

**Authors:** Gregory Christodoulidis, Athina A. Samara, Alexandros Diamantis, Theodoros Floros, Ioanna-Konstantina Sgantzou, Kostas-Sotirios Karakantas, Prokopis-Andreas Zotos, Antonios Koutras, Michel B. Janho, Konstantinos Tepetes

**Affiliations:** Department of General Surgery, General University Hospital of Larissa, Mezourlo, 41110 Larissa, Greece; gregsurg@yahoo.gr (G.C.); alexandrosdoc@gmail.com (A.D.); ftheodor93@gmail.com (T.F.); iksgantzou@gmail.com (I.-K.S.); geokara1983@hotmail.com (K.-S.K.); zotospro@hotmail.com (P.-A.Z.); antoniskoy@yahoo.gr (A.K.); Micheljanho@live.co.uk (M.B.J.); tepetesk@gmail.com (K.T.)

**Keywords:** *Echinococcus*, liver, complications, endemic, hydatid disease

## Abstract

*Background and Objectives:* Hydatid disease (HD) remains a significant public health issue causing morbidity and mortality in many Mediterranean countries. *Material and Methods:* The present cohort study included 50 consecutive patients with liver hydatid disease who underwent surgery in a tertiary University Hospital. A total of 18 patients (36%) had a case of complicated HD, including simple communication of the cyst with the biliary tree (6 cases), rupture of the cyst into the biliary tree (6 cases), presence of a bronco-biliary fistula (2 cases), rupture of the cyst in the peritoneal cavity (2 cases), and rupture of the cyst and formation of a hepatic abscess (2 cases). Endoscopic retrograde cholangiopancreatography (ERCP) was pre-operatively performed on six patients. *Results:* The main clinical symptom presented was right upper quadrant pain in 16 patients (88%), which was associated with high fever (>39 °C) in 14 patients (78%). C-reactive protein (CRP) was the primary indicator of a complicated HD (*p* = 0.003); however, it was only elevated in 67% of cases. CRP was a more sensitive indicator of a rupture in the biliary tree cyst (*p* = 0.02). Computer tomography (CT) detected more cases (44%) of a complicated HD than ultrasonography (US) (25%); however, the difference was not statistically significant. *Conclusions:* For prevention and control of HD, a high suspicion of the disease leading to early referral to specialized centers, mainly in endemic areas, is required. Prior to surgical or percutaneous intervention, a combination of imaging and laboratory findings are essential in diagnosing a complicated case and avoiding unnecessary interventions.

## 1. Introduction

Hydatid disease (HD) or echinococcosis is a chronic parasitic disease caused by the larval stage of the tapeworm *Echinococcus granulosus*. To date, it remains a severe public health issue with a considerable economic impact. This disease contributes to morbidity and mortality in many parts of the world, primarily Mediterranean countries, the Middle East, New Zealand, Australia, India, and South America, due to the close association between sheep, dogs, and humans [1,2,3,4,5]. Humans are considered intermediate hosts that become infected by *Echinococcus* inadvertently. Intermediate hosts are infected by ingesting eggs within the faeces of the definite hosts. The eggs liberate their larvae into the duodenum, crossing the intestinal wall and, via the portal system, reach the liver, where they develop into cysts [6,7,8,9].

HD can involve almost every part of the body; however, the liver is the most commonly affected organ (75%), followed by the lungs (15%) and other locations, such as the brain (2%) or the spine (1%) [1,3,10]. Without treatment, hydatic cysts gradually grow and can develop fistulas with adjacent organs or the biliary tree. The hydatic cysts may also rupture into the peritoneal cavity causing an anaphylactic reaction and the seeding of secondary cysts, develop daughter cysts within, or die [10]. The rupture of a hydatid cyst can occur in 50–90% of cases, leading to an allergic response due to the presence of antigenic cyst content in the surrounding structures. These clinical syndromes may lead to a patient’s death, but treatment does not ensure improvement of the patient’s quality of life [11,12,13,14,15].

The aim of the present study is to evaluate the diagnostic importance of several biochemical and radiological values in the diagnosis of complicated liver hydatid disease.

## 2. Materials and Methods

### 2.1. Study Design—Settings

All consecutive patients who were admitted to a tertiary University Hospital diagnosed with hydatid liver disease during a 7-year period (between 2009 and 2016) were included in the present study. The Strengthening the Reporting of Observational Studies in Epidemiology (STROBE) checklist was used [16]. Ethical review and approval were waived for this study, due to the retrospective study design based primarily on medical records and laboratory results, as well as the anonymity of patients in the statistical analysis.

#### 2.1.1. Variables of the Participants

In this retrospective analysis, we reviewed the medical records of 50 patients that underwent surgery due to echinococcal liver disease. A total of 18 (36%) patients were presented as a complicated case, with the final diagnosis set intra-operatively.

Parameters considered for the evaluation of these patients included the following: age, gender, co-morbidities, clinical presentation, physical examination, laboratory findings, imaging modalities (such as ultrasonography (US), computer tomography (CT), magnetic resonance (MRI), and other interventions, including Endoscopic Retrograde Cholangiopancreatography (ECRP) or Magnetic Resonance Cholangiopancreatography (MRCP)), duration of hospitalization, complications, mortality, type of surgery performed, and follow-up. Laboratory findings included serum leucocyte and eosinophile count, C-reactive protein (CRP), aspartate and alanine transaminases (AST and ALT, respectively), γ-glutamic transaminase (γGT), alkaline phosphatase (ALP), total and direct bilirubin (TB, DB respectively), and anti-echinococcal antibodies. Data regarding the number, size (maximum diameter), and location of the cyst were also evaluated.

Communication or rupture of the cyst within the biliary tree was suspected pre-operatively due to clinical presentation (i.e., obstructive jaundice or acute cholangitis), laboratory findings (i.e., increased levels of bilirubin or cholestatic enzymes), and/or positive imaging findings (i.e., dilatation of common biliary duct with or without the presence of material within it). A broncho-biliary fistula was suspected due to the presence of yellowish sputum and dyspnea on an otherwise fit and well patient. However, in all 18 cases, the final diagnosis was set intra-operatively and was confirmed by post-operative histological examination.

#### 2.1.2. Statistical Analysis

Data were processed using SPSS V26 (IBM, Chicago, IL, USA). Qualitative variables are presented as absolute (N) and relative (%) frequencies, and quantitative variables are presented as means with standard deviation (SD). Statistical analysis for categorical variables was performed using χ^2^ and Fischer exacts test, and student’s *t*-test for descriptive variables. A *p*-value of <0.05 was considered statistically significant.

## 3. Results

A total of 18 of the 50 (36%) patients that underwent surgery for a hydatic cyst located in the liver were diagnosed with a complicated hydatic cyst and were included in the present study. The patients’ demographic and basic characteristics are displayed in Table 1. The mean patient age was 56.3 years (range 25–82 years); 14 (78%) patients were male and 4 (22%) were female. In 12 cases (67%), there was a communication of the cyst with the biliary tree, either as simple a communication of the cyst with the biliary tree (CBT, 6 cases) or as a rupture of the cyst into the biliary tree (RBT, 6 cases). There were 2 cases of bronco-biliary fistula (BBF, 11%), 2 cases of a cyst rupture in the peritoneal cavity (RPC, 11%), and 2 cases of cyst rupture and formation of a hepatic abscess (11%).

Furthermore, the mean number of cysts was 1.3 (range 1–4) with a mean maximum diameter of 9.6 cm (range 4.5–20 cm). In one case, the cyst was enormous and occupied the whole right lobe and the left medial part of the liver (Figure 1), while in another case a large right liver lobe cyst compressed the hilum resulting in left liver lobe biliary dilatation (Figure 2).

It is noteworthy that in 9 cases (50%), the diagnosis of a complicated liver hydatid cyst was unclear at the time of referral. One case of broncho-biliary communication was initially regarded as an exacerbation of Chronic Obstructive Pulmonary Disease (COPD), while two further cases were considered as an exacerbation of tuberculosis. A total of 4 patients (22%) had previously received medical treatment for liver hydatid disease, but had missed their follow-up. Moreover, two cases were misdiagnosed as Klatskin or adrenal tumor.

The main clinical symptom presented was right upper quadrant pain in 16 patients (88%), which was associated with high fever (>39 °C) in 14 patients (78%). A total of 2 patients (16%) with involvement of the biliary tree presented with the typical signs of acute cholangitis, whereas both patients with BBF presented with yellowish sputum along with upper right quadrant pain and fever. One patient with RPC presented with acute abdomen, without specific signs of hydatid disease. All patients had non-typical symptomatology (Table 1).

Table 2 presents the number of cases with increased biochemical values, classified by the type of complicated hydatic disease. Anti-echinococcal antibodies were measured in only 5 patients (28%) and were positive in 4 of them (80% of measured cases). When comparing the main biochemistry examination, CRP was the main indicator of a complicated HD (*p* = 0.003) as it was elevated in 12 cases (67%), followed by ALP/γGT that was elevated in 11 patients (61%), eosinophils count that increased in 10 cases (56%), and total bilirubin levels that were elevated in only 2 cases (11%) of rupture in the biliary tree. Furthermore, CRP was a more sensitive indicator of a rupture in the biliary tree cyst (increased in 5 cases, 83%) when compared to the increase in cases of simple communication with the biliary tree (increased in only 1 case, 17%) (*p* = 0.02).

Table 3 summarizes the diagnostic value of preoperative image modalities classified by definitive intra-operative diagnosis. More specifically, an ultrasound scan was performed in 16 patients (89%) (excluding patients with a broncho-biliary fistula) and was diagnostic of intrabiliary rupture in 2 cases, which was 16% of total cases that involved the biliary tree and 33% of all cases with a cyst ruptured into the biliary tree. Ultrasound scanning was also diagnostic in 2 CBT cases (33%). Overall, the ultrasound scan was diagnostic in 33% of cases with a cyst involved in the biliary tree (4/12 patients). An ultrasound scan was also performed in RPC patients, with no diagnostic value; therefore, an abdominal CT scan was performed to establish the diagnosis.

A CT scan was performed in all (18) complicated cases, of which 8 cases (44%) were diagnosed with complicated HD. More precisely, the CT scan was diagnostic in 2 RBT cases (33%). It should be noted that, in one case, the hydatid mass was reported as an adrenal mass. CT scanning also failed to identify the simple communication of the cyst with the biliary tree, as it was non-diagnostic in 5 out of 6 cases (83%). For BBF and RPC cases, a CT scan was diagnostic in all cases, which, apart from the hepatic lesion, showed the diaphragmatic erosion, the infiltration of the lung and the bronchi, and the presence of pericystic abdominal fluid. In cases of hepatic abscess, a CT scan was diagnostic in one case only, presenting air within the cyst. Dilatation of the bile ducts were observed in 2 cases (11%) and paracystic fluid collection was present in all cases of complicated HD and hydrothorax, in both cases with a broncho-biliary fistula.

Additional imaging modalities included MRI/MRCP. These modalities were performed as a means of further investigation when US and CT could not lead to a definitive diagnosis in two cases of RBT. In both cases (100%), a definitive diagnosis of a common bile duct cyst was set with MRI/MRCP.

When comparing the diagnostic accuracy of CT and US, CT detected a greater number of complicated HD cases (8/18, 44%) than US (4/16, 25%). However, the difference was not statistically significant (*p* = 0.2). A statistically significant difference (*p* = 0.045) was found in cases in which the cyst was ruptured in the peritoneal (PRC), with CT being diagnostic in all cases (2/2) and US diagnostic in none (0/2). Moreover, CT detected a case of AF (50%), while US could detect none (0/2), but the difference was not statistically significant (*p* = 0.08).

Pre-operative ERCP was performed in a total of 6 cases (33%). This procedure was conducted in both cases of BBF and in the four cases in which a rupture was highly suspected by imaging findings.

## 4. Discussion

Hydatid disease remains a serious public health challenge contributing to morbidity and mortality worldwide, particularly in Mediterranean countries where the disease is endemic. In these regions, the annual incidence of HD ranges from less than 1 to 200 per 100,000 inhabitants [8]. Greece is a highly endemic area for echinococcosis, with an annual incidence of 0.3 per 100,000 inhabitants [15,17]. Echinococcosis is not only a severe disease that may result in death (mortality rate 1–2%), but also a disease that can lead to severe economic losses due to treatment costs, lost wages, and losses in the livestock industry. It has been estimated that disability-adjusted life year (DALY) losses are approximately US $193,529,740 globally [18,19].

Unexpectedly, over a 7-year period, only 50 cases of hydatid disease were referred to our University hospital, despite the hospital serving a largely rural population of approximately 700,000. Eighteen of these patients presented to our unit as complicated cases of hydatid disease. Their delayed presentation at the University hospital can be attributed to either an underestimation by primary healthcare physicians of initial presenting symptoms, or failure on the part of the patients to follow medical advice received. This highlights both the low level of suspicion and the population’s lack of knowledge related to the disease, despite the region of Thessaly being a leading Mediterranean endemic area. Public education/awareness campaigns should be developed to inform and educate populations living in endemic areas for the early recognition of HD symptoms and to seek timely medical advice. Furthermore, for prevention and control of the disease the World Health Organization (WHO) suggests periodic dog deworming, improving hygienic conditions in slaughterhouses, and the vaccination of livestock with an *E. granulosus* recombinant antigen (EG95) [15].

Liver hydatid disease is frequently silent, with approximately 75% of patients experiencing an asymptomatic abdominal mass incidentally diagnosed during abdominal investigation for another pathology. In symptomatic cases, the most common symptom displayed is epigastric pain or pain located in the right upper abdominal quadrant, and typical findings include an enlarged liver and a palpable mass. In many cases, patients seek medical advice at a late stage when complications are already established. These complications include rupture in the biliary tree, infection, rupture into viscera, intraperitoneal or intrathoracic rupture, and external compression leading to portal hypertension [7,10,20]. Rupture can occur because of trauma or relentless expansion of the cyst resulting rupture into the peritoneal cavity, the pleural cavity or the bile duct. When located in the superior and posterior liver segments, HD can grow upwards towards the chest, eroding through the diaphragm and resulting in fever, cough, and bile-stained sputum. Rupture into the peritoneal cavity is accompanied by shock and signs of diffuse peritonitis [10]. Laboratory findings can be useful in the diagnosis of liver hydatid disease, although they are not always conclusive as indicators for biliary tree involvement. Differential diagnosis of simple communication from rupture into the biliary tree cannot rely on bilirubin levels or other liver tests [4]. Moreover, the use of the anti-echinococcal antibodies in diagnosing the disease remains unclear [21]. In our series, anti-echinococcal antibodies were tested in only five patients with a complicated disease. The eosinophile count was slightly elevated in most complicated cases, but failed to differentiate rupture from simple communication. CRP provided a diagnostic indicator in RBT cases. This may be due to the fact that the rupture of the cyst is a prerequisite for the contents to be released into the biliary tree, and the rupture results in an inflammatory reaction. Furthermore, Gram negative bacteria may infect the ruptured cyst causing an inflammatory reaction, which is unlikely to happen when simple communication occurs. In our study, CRP was the main indicator of a complicated HD (*p* = 0.003); however, it was elevated in only 67% of cases. Therefore, without a specific blood test for the diagnosis of liver HD, a wide range of biochemical values should be investigated to support the diagnosis [22,23].

Radiology imaging is the main diagnostic tool for the diagnosis of HD. This tool can detect the disease even in asymptomatic cases, or cases with mild and non-specific symptoms, which may not be easily evaluated via clinical examination. Ultrasonography and a CT scan are typically used in diagnosis of liver cystic lesions. In our study, CT detected more cases (44%) of a complicated HD than US (25%); however, the difference was not statistically significant. In cases of broncho-biliary fistula and free intraperitoneal rupture, CT was proven to be adequate for diagnosis; however, it was inconclusive in terms of biliary tree involvement. Moreover, CT offers a better view of the hydatid cysts regarding location and relationship with surrounding structures, while US is a non-invasive, cost-effective method that can be used as an initial diagnostic tool for cystic lesions of the liver. HD cysts appear as lobulated structures containing daughter vesicles or membranes and septa, while the wall may appear partially or heavily calcified [2,7,20]. Furthermore, our experience indicates that a preoperative MRI/MRCP should not be performed as a routine, but in cases with an obvious communication with the biliary tree. Prior to a surgical or percutaneous intervention, a combination of imaging and laboratory findings are essential in diagnosing a complicated case and avoiding unnecessary interventions.

Endoscopic retrograde cholangiopancreatography (ERCP) remains an important diagnostic tool in cases of major biliary communication, while simultaneously allowing the clearance of the common bile duct and reducing the risk of a post-operative fistula [7,24]. Many authors suggest that ERCP should be performed in every case with clinical or biochemical suspicion of biliary involvement, before or after any surgical procedure [24]. ERCP is a safe and effective procedure in means of avoiding strictures, removal of endocystic material, closure of fistulas, and reduction of the possibility of another operation [25,26]. The majority of our cases included involvement of the biliary tree, which was indicated by both biochemical and imaging findings. In our study, pre-operative ERCP was performed in 6 cases (33%) of biliary communication, with favorable results for the patients.

The most common complication in the management of hydatid disease is cyst rupture into the biliary tree or the thoracic cavity. Biliary communications can occur with a frequency ranging from 3.5% to 19% [2,11]. Less common complications involve broncho-biliary fistulas and a rupture of the cyst in the peritoneum. Broncho-biliary fistula may present with a simple cough, bile-stained sputum, or even hydatidemesia [27,28]. A rupture of the cyst in the abdomen, which result from trauma or high intra-abdominal pressure, is a severe situation and should always be treated vigorously with emergency laparotomy [29,30].

To the best of our knowledge, the present study identified the diagnostic value of several biochemical markers and evaluated radiological modalities to diagnose complicated liver hydatid disease for the first time in the literature. However, prior to the appraisal of these results, several limitations should be considered. Data were collected retrospectively depending on the availability and accuracy of the data records, and information bias may have occurred. Due to the limited number of patients, the results of the statistical analysis should be interpreted cautiously.

## 5. Conclusions

In conclusion, echinococcosis remains a serious public health issue leading to impaired quality of life for affected people and even death. Health providers should be highly suspicious of liver hydatid disease in endemic areas, and should refer any suspected case to a specialized center for definitive diagnosis and treatment. The diagnosis of a complicated liver HD can be established through clinical, biochemical (mainly with CRP), and imaging findings (US and CT).

## Figures and Tables

**Figure 1 medicina-57-01210-f001:**
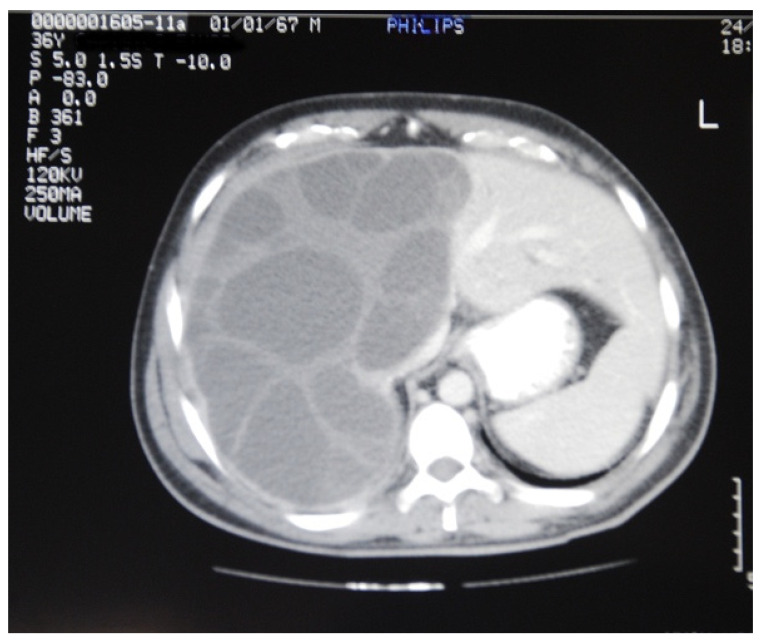
An enormous hydatid cyst occupying the whole right lobe and the left medial part of the liver.

**Figure 2 medicina-57-01210-f002:**
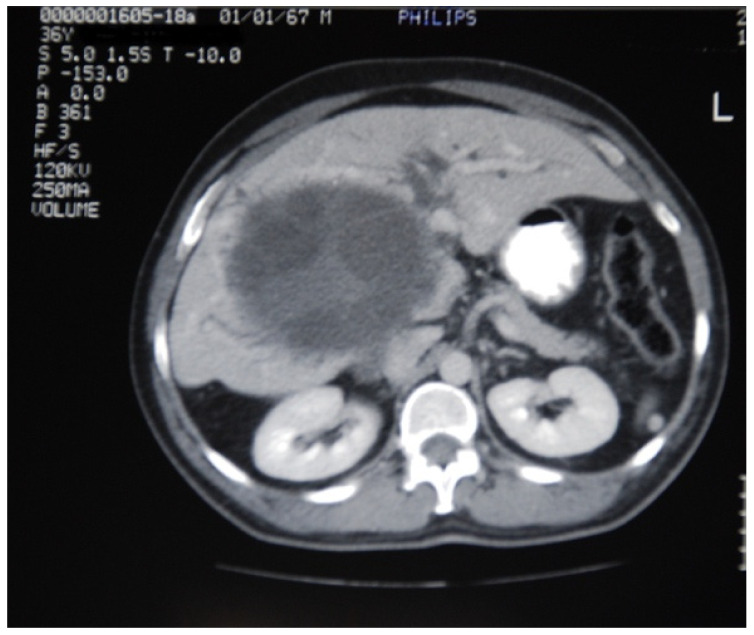
A large hydatid cyst of the right lobe of liver compressing the hilum, resulting in left liver lobe biliary dilatation.

**Table 1 medicina-57-01210-t001:** Basic characteristics of patients regarding co-morbidities, clinical presentation and cyst characteristics.

Value/Group	Complicated HD (18) N (%)	Uncomplicated HD (32) N (%)	*p*-Value
**Age** (mean, range)	56.3 (25–82)	55.1 (23–81)	0.875
**Male gender**	14 (78%)	26 (81%)	0.479
**Previous medical advice**	9 (50%)	17 (53%)	0.831
Missed diagnosis	7 (39%)	10 (31%)	
Known hydatid disease	2 (11%)	7 (22%)	
**Co-morbidities**	13 (72%)	20 (63%)	0.486
Cardiovascular	5 (28%)	10 (31%)	
Respiratory	4 (22%)	7 (22%)	
Diabetes mellitus	2 (11%)	3 (9%)	
>1 co-morbidities	6 (33%)	12 (60%)	
**Clinical presentation**			
Non-specific	18 (100%)	29 (91%)	
Nausea, vomiting	17 (94%)	22 (69%)	
RUQ tenderness	16 (88%)	25 (78%)	
Fever	14 (78%)	7 (22%)	
Yellow sputum	2 (11%)	0	
Acute cholangitis	2 (11%)	0	
Acute abdomen	1 (0.5%)	0	
**Cyst characteristics**			
Number of cysts (mean, range)	1.3 (1–4)	1 (1–3)	0.148
Maximum diameter (mean, range)	9.6 (4.5–20)	7.6 (4–18)	0.097

**Table 2 medicina-57-01210-t002:** Cases of elevated biochemical values classified by definitive intra-operative diagnosis.

Elevated Biochemical Values	Intra-Operative Diagnosis (N%)					
	Biliary Tree Rupture (6)	Biliary Tree Communication (6)	Broncho-Biliary Fistula (2)	Rupture in the Peritoneal Cavity (2)	Abscess Formation (2)	Total (18)
Bilirubin	2/6 (33%)	0/6 (0%)	0/2 (0%)	0/2 (0%)	0/2 (0%)	2/18 (11%)
ALP/γGT	4/6 (66%)	4/6 (66%)	2/2 (100%)	1/2 (50%)	0/2 (0%)	11/18 (61%)
CRP	5/6 (83%)	1/6 (17%)	2/2 (100%)	2/2 (100%)	2/2 (100%)	12/18 (67%)
Eosinophiles	2/6 (33%)	2/6 (33%)	2/2 (100%)	2/2 (100%)	2/2 (100%)	10/18 (56%)

**Table 3 medicina-57-01210-t003:** Diagnostic value of preoperative image modalities classified by definitive intra-operative diagnosis.

	Intra-Operative Diagnosis (N%)					
	Biliary Tree Rupture (6)	Biliary Tree Communication (6)	Broncho-Biliary Fistula (2)	Rupture in the Peritoneal Cavity (2)	Abscess Formation (2)	Total (18)
Ultrasonography	2/6 (33%)	2/6 (33%)	Not performed	0/2 (0%)	0/2 (0%)	4/16 (25%)
Computer Tomography	2/6 (33%)	1/6 (17%)	2/2 (100%)	2/2 (100%)	1/2 (50%)	8/18 (44%)
Magnetic Resonance Imaging	2 (100%)	Not performed	Not performed	Not performed	Not performed	2/2 (100%)

## Data Availability

Data are available upon reasonable request.

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
