# Peer review of "Reaching the Challenging Diagnosis of Complicated Liver Hydatid Disease: A Single Institution’s Experience from an Endemic Area"

_medicina, 2021, doi:10.3390/medicina57111210_

Round 1
Reviewer 1 Report
The paper deserves to be published because:
Liver disease and especially liver disease is a public health problem.
Due to the mobility of the population, it must be known by all doctors.
Prophylaxis of the disease is important
Complications are relatively common.
Hepatic hydatid cyst should be treated in experienced surgical centers (volume centers).
Author Response
We woulk like to thank the reviwer for his kind comments.
Reviewer 2 Report
The article presents the data of a single-center study on the relevant issue of diagnosing complicated forms of liver echinococcosis. The authors came to the conclusion that there are no reliable criteria for preoperative diagnosis of a complicated course of hepatic echinococcosis, with the exception of determining the CRP level.
The data presented by the authors are very interesting, but from the point of view of the evidence and depth of data analysis, the article requires correction.
Below is a list of comments
- the article contains a lot of unnecessary information about the well-known data on the pathogenesis of the disease and an obvious shortcoming about the mechanisms and factors of the complicated course of liver cysts
- The authors did not use the WHO classification, and therefore there is no understanding which types of cysts were more often associated with a complicated course and how this factor relates to the risk of complications
- Table 1 provides a simple listing of means for demographic data, clinical presentation, and also size and number of cysts. It would be better to compare these data between the groups of complicated and uncomplicated course of the disease.
- It was also important to include other characteristics, such as the presence of dilatation of the bile ducts, paracystic fluid collection (including the presence of sympathetic hydrothorax).
- It would be important to analyze the impact of disease duration as a risk factor, although this data may not be available in all patients.
- In the summarizing part of the article, there is no generalization of the research results, although it is obvious that the authors' data indicate the presence of clinical and laboratory data indicating an acute inflammatory response (hyperthermia and increased CRP)
- the discussion does not compare the data obtained by the authors with other studies outcomes
- There is no section describing the limitations of the study
The article requires correction and repeat reviewing
Author Response
Reviewer’s concern: The article presents the data of a single-center study on the relevant issue of diagnosing complicated forms of liver echinococcosis. The authors came to the conclusion that there are no reliable criteria for preoperative diagnosis of a complicated course of hepatic echinococcosis, with the exception of determining the CRP level.
The data presented by the authors are very interesting, but from the point of view of the evidence and depth of data analysis, the article requires correction.
Authors’ answer: After thanking the reviewer about the comments we would like to answer separately to every very well addressed concerns.
Reviewer’s concern: the article contains a lot of unnecessary information about the well-known data on the pathogenesis of the disease and an obvious shortcoming about the mechanisms and factors of the complicated course of liver cysts
Authors’ answer: We have modified introduction and discussion sections according to the present well stated comment focusing on mechanisms of a complicated liver cyst.
Reviewer’s concern: The authors did not use the WHO classification, and therefore there is no understanding which types of cysts were more often associated with a complicated course and how this factor relates to the risk of complications.
Authors’ answer: Because the Ultrasonography examination was taken place under emergency settings, WHO classification was not available.
Reviewer’s concern: Table 1 provides a simple listing of means for demographic data, clinical presentation, and also size and number of cysts. It would be better to compare these data between the groups of complicated and uncomplicated course of the disease.
Authors’ answer: Table 1 was modified accordingly.
Reviewer’s concern: It was also important to include other characteristics, such as the presence of dilatation of the bile ducts, paracystic fluid collection (including the presence of sympathetic hydrothorax).
Authors’ answer: Dilatation of the bile ducts were observed in 2 cases (11%) and paracystic fluid collection were presented in all cases of complicated HD and hydrothorax in both cases with a bron-chobilary fistula. This statement was added in the results section.
Reviewer’s concern: It would be important to analyze the impact of disease duration as a risk factor, although this data may not be available in all patients.
Authors’ answer: This is a well-stated comment, but unfortunately due to the retrospective nature of the study, these data are not available.
Reviewer’s concern: In the summarizing part of the article, there is no generalization of the research results, although it is obvious that the authors' data indicate the presence of clinical and laboratory data indicating an acute inflammatory response (hyperthermia and increased CRP)
Authors’ answer: Thank you for the well-stated comment. The conclusion section was modified accordingly.
Reviewer’s concern: the discussion does not compare the data obtained by the authors with other studies outcomes
Authors’ answer: To the best of our knowledge, the present study has identified the diagnostic value of several biochemical markers and evaluated radiological modalities in diagnosing complicated liver hydatid disease for the first time in literature
Reviewer’s concern: There is no section describing the limitations of the study
Authors’ answer: A new paragraph was added in the discussion section analyzing strengths and limitation of the present study.
Reviewer 3 Report
The reviewer would like to thank the authors for their nice attempt in dealing with such a hot issue, especially in Mediterranean countries.
Some clarifications, however, would be of benefit. In more detail, it would interesting that the authors expain/comment on:
- The diagnostic value of the combination of anti-echinococcal antibodies in combination with ultrasound or computed tomography.
- Why only 6 and not all 12 patients with either biliary tree rupture or biliary tree communication underwent a preoperative ERCP?
- Is authors' opinion that a preoperative routine MRI/MRCP imaging might have altered the operative approach?
Author Response
Reviewer’s concern: The reviewer would like to thank the authors for their nice attempt in dealing with such a hot issue, especially in Mediterranean countries. Some clarifications, however, would be of benefit. In more detail, it would interesting that the authors expain/comment on
Authors’ answer: After thanking the reviewer about the comments we would like to answer separately to every very well addressed concerns.
Reviewer’s concern: The diagnostic value of the combination of anti-echinococcal antibodies in combination with ultrasound or computed tomography.
Authors’ answer: Unfortunately, the laboratory of microbiology/ immunology had a limited source of resources for measuring anti-echinococcal antibodies and they were noty available for a long time during study period. Having these limitations, anti-echinococcal antibodies were not measured in all cases (they were available only in 5 from the 18 cases) and a combined diagnostic value can not be calculated.
Reviewer’s concern: Why only 6 and not all 12 patients with either biliary tree rupture or biliary tree communication underwent a preoperative ERCP?
Authors’ answer: A preoperative ERCP was conducted in both cases of BBF and in 4 cases where a rupture was highly suspected by imaging findings. In most of the other cases the diagnosis of biliary tree rupture or communication was set intraoperatively.
Reviewer’s concern: Is authors' opinion that a preoperative routine MRI/MRCP imaging might have altered the operative approach?
Authors’ answer: Our experience indicates that a preoperative MRI/MRCP should not be performed as a routine, but in cases with an obvious communication with the biliary tree. This sentence has been added in discussion section.
Round 2
Reviewer 2 Report
Since the study is retrospectively designed and analyzes data from emergency diagnosis and treatment, many of the limitations cannot be overcome. At the same time, the subject of study and the results obtained are of undoubted interests . The paper can be accepted for publication after minimal stylistic corrections
Author Response
We would like to thank again the reviwer for his/her kind and helpful comments. Unfortunatelly, due to the relative rarity of the presentation of a complicated hydatid liver cyst, data design were collected retrospectively during a 7-year period. Due to this retrospective design of the study and the diagnosis in emergency setting, many limitations were psesented as they were analyzed in a paragraph in the discussion section after the reviewer's well-stated comment.
Moreover, minimal stylistic corrections were applied in the new version submitted.